# Molecular Screening of the Thrombophilic Variants Performed at G-141 Laboratory among Saudi Infertile Women

**Arwa A. Alageel [1], Maysoon Abdulhadi Alhaizan [2], Salwa Mohamed Neyazi [2], Malak Mohammed Al-Hakeem [2] and Imran Ali Khan [1],***

[1] Department of Clinical Laboratory Sciences, College of Applied Medical Sciences, King Saud University, Riyadh 11433, Saudi Arabia; aaalageel@ksu.edu.sa
[2] Department of Obstetrics and Gynecology, College of Medicine, King Khalid University Hospital, King Saud University, Riyadh 11451, Saudi Arabia; malhaizan@ksu.edu.sa (M.A.A.); sneyazi@ksu.edu.sa (S.M.N.); malhakeem@ksu.edu.sa (M.M.A.-H.)
* Correspondence: imkhan@ksu.edu.sa; Tel.: +966-501112806

**Abstract:** Infertility is a major issue at present and is a common disease that exists in both male and female reproductive systems, described as failure to attain pregnancy. The most important physiological phenomenon for establishing clinical pharmacy is defined as female infertility (FI). Obesity enhances the risks for many chronic disorders, especially causing a high risk for women's reproductive health. The relationship between infertile women and thrombophilia is characterized by abnormal blood coagulation. Among the thrombophilic variants, Factor V Leiden (FVL), prothrombin (PT) and methyl tetrahydrofolate reductase (MTHFR) in genes such as G1691A (rs6020), G20210A (rs1799963) and C677T (rs1801133) are commonly studied in the majority of human diseases. In this case–control study, we investigated the role of thrombophilic variants such as G1691A, G20210A and C677T in the *FVL*, *PII* and *MTHFR* genes in Saudi infertile women. Based on sample size calculation, 100 female infertile and 100 control (fertile) women were selected based on inclusion and exclusion criteria. Genotyping was performed with polymerase chain reaction and followed with precise restriction enzymes, which can accurately detect the nucleotide amendment variants in G1691A, G20210A and C677T. The required statistics were applied between the case (infertile) and control (fertile) women to document the role of the G1691A, G20210A and C677T variants in Saudi infertile women. In this study, age, weight and BMI were found to be high in the control women in comparison to the infertile women. None of the genotypes, genetic models or allele frequencies were associated with G1691A, G20210A or C677T SNPs ($p > 0.05$). Furthermore, the regression model and ANOVA analysis also showed negative statistical associations. The combination of genotypes and allele frequencies among G1691A, G20210A and C677T SNPs showed positive associations in the recessive model ($p = 0.0006$). Finally, the GMDR model showed moderate associations with the gene–gene interaction, dendrogram and depletion models. Finally, this study confirmed that thrombophilic SNPs have no role and may not be involved in Saudi infertile women.

**Keywords:** female infertility; thrombophilia; G1691A; G20210A; C677T variants and Saudi women





## 1. Introduction

Infertility is a human disease of the male and female reproductive systems, and according to the World Health Organization (WHO), infertility affects around 15% of couples of reproductive age [1]. The American Society for Reproductive Medicine defines infertility as being characterized by the inability to conceive after having unvarying and unguarded vaginal sex for a minimum of 365 days in a year [2], which affects around 48.5 million couples globally [3]; however, it varies by ethnic region and 8–12% is an estimation of the infertility among reproductive-aged couples [4]. Pelvic adhesions, endometriosis, ovulatory disorders, hyperprolactinemia and tubal anomalies are considered the female factors and

account for 35% of infertility cases, while male factors are responsible for 30% and 20% is based on a combination of factors from both the genders. Idiopathic or unexplained infertility refers to unknown causes of infertility, present in 15% of cases [5]. Infertility is characterized into (i) primary and (ii) secondary infertilities, in which the primary infertility is aligned with the aforementioned definition of infertility, and secondary infertility involves extended efforts by couples to conceive and can either lead to a successful pregnancy or results in miscarriage [6]. One of the common causes of infertility in couples could be dysfunction of the reproductive system [7]. Globally, the prevalence of infertility is expanding at an alarming rate, resulting in physical and mental health effects in infertile couples, along with other negative impacts in their lives. The prevalence of female infertility (FI) has been reported to be between 33% and 41% globally [8]. FI is described as a complex disorder caused by anatomical, autoimmune, hormonal, genetic, thrombotic and unidentified infectious factors. Between 5% and 10% of infertile women have genetic abnormalities, which could be chromosomal anomalies, deletions, duplications, aberrations, insertions, mutations or polymorphisms in origin [9]. FI is associated with overweight/obesity, diabetes, polycystic ovary syndrome (PCOS), fibroids, thyroid disorders, endometriosis, anemia and eating disorders [10]. In the present era, the prevalence of obesity is expanding in Saudi Arabia, especially in Saudi women compared to Saudi men. Reports by the WHO have estimated overweight and obesity in women and men in the Saudi Arabia to be around 69.2%, 39.5% and 67.5%, 29.5%, respectively [11]. Obesity involves an expansion of adipose tissue due to the hypertrophy of adipose cells which store triglycerides (TG) [12]. The first case of obesity and infertility was documented in 400 BC in a Scythian tribe, which was defined as either overweight or obesity [13]. Maternal obesity causes an elevated body mass index (BMI), another risk factor that adversely affects FI. The predicted incidence of maternal obesity is expected to increase from 45–50% till 2030 according to the World Obesity Federation [14]. The fertility rate decreases in accordance with increasing obesity; women with a BMI > 30 kg/m$^2$ have a 2.7-fold amplified risk of infertility, as well as around a 25–37% chances of miscarriage. Additionally, the treatment for infertility in obese women and its response rate is low and has a high risk of early miscarriage after IVF treatment, while the live birth rate has dropped to 20% [15]. PCOS, thyroid problems, elder age, weight gain, diabetes and high blood pressure are common causes seen in infertile women [16]. Conception and pregnancy complications are interrelated to variants of thrombophilia [9,17,18] and can also be defined as the abnormality of blood coagulation, which elevates the risk of thrombosis [19]; this is demarcated as an expanding tendency in thrombosis via exacerbated coagulation. The susceptibility of individuals to thrombotic diseases owing to hemostatic system dysregulation is influenced by various inherited and lifestyle-related factors in thrombophilia. Increased thrombosis, especially venous thrombosis, was originally attributed to inherited deficits of antithrombin and protein C/S, which are anticoagulation-related factors [20,21]. Thrombophilia is a medical condition characterized by an increased tendency to form blood clots, which can lead to a higher risk of thrombosis, which is the formation of abnormal blood clots within blood vessels. Blood clots are a normal part of the body's response to injury and play a crucial role in preventing excessive bleeding. However, in individuals with thrombophilia, the blood clotting process can become overactive, leading to the formation of clots inappropriately or more easily [22]. Both the congenital and acquired forms of thrombophilia are related to modifications in hemostatic mechanisms, characterized via an enhanced tendency toward blood clotting and a risk of consequences in thromboembolism. The predisposition to thrombosis may be related to molecular factors, such as (i) the G1691A variant in Factor V Leiden (FVL), (ii) the G20210A variant in prothrombin (PRT/FII) and (iii) the C677T variant in methyl tetrahydrofolate reductase (MTHFR) genes [23]. The G1691A and C677T SNPs are present in the exonic region, and the G20210A SNP is present in the 3′UTR region. Thrombophilic variants in FI can overactivate the process of coagulation, which further affects the homocysteine metabolism or causes the process of methylation deregulation. Genetic variants in thrombophilia are suspected, as 30% of obstetric complications are associated [24]. One of the risk

factors for thromboembolic events is inherited thrombophilia [22], and previous studies have shown a relationship between infertile women [22] and pregnancy-related issues such as recurrent pregnancy loss (RPL) [25], complications with in vitro fertilization [26], implantation failures [27] and ovarian hyperstimulation syndrome [28]. The alternative term for FI or infertile women is reproductive failure (RF), which is described as an inability to conceive or carry a pregnancy. Infertile women have been found to have a 50% higher rate of thrombophilia [29]. Obesity can be considered a negative impact on human health, particularly in infertile women or those undergoing fertility treatment [30]. Previous cohort studies have consistently reported the role of obesity as leading to abortion in women who underwent IVF [31]. The prevalence of obesity is high in Saudi Arabia [32], and obesity can be considered central to the development of all human diseases, particularly metabolic diseases and reproductive diseases in women [33–35]. Female obesity has a great impact on reproductive function and the hormonal milieu [36]. In women, obesity is linked to puberty, menstrual issues and PCOS during childhood and adolescence [37]. Previous studies have found inconsistent results between childhood obesity and FI [38,39]. Another previous study on childhood obesity, which was followed up for 25 years, showed the development of FI during reproductive age [40].

Thrombophilia is commonly connected with venous thromboembolism and is considered a multifactorial disease affected by environmental and inherited risk factors, i.e., genetic variants in the *FVL*, *PT* and *MTHFR* genes [41]. The hematological changes associated with enhanced blood coagulability and inclination in the direction of vascular thrombosis are triggered by inherited thrombophilia [42]. The genetic substrate of the prothrombotic state can either be a gain or loss of functional disorders, in which the *FVL/PT/MTHFR* genes are considered a gain of functional disorders, and proteins C/S and antithrombin III (AT III) are considered a loss of functional disorders [43]. FVL refers to a point mutation at nucleotide 1691 of exon 10 of the *FVL* gene (G to A transition) that results in a procoagulant condition, owing to the loss of one of the activated protein C (APC) cleavage sites. The amino acid substitutes guanine to adenine via CGA–CAA, and this leads to the missense variant at the 506th position [44]. The alternative name for the prothrombin (PT) gene is factor II (FII), which is a well-known precursor of thrombin. The *PT* gene functions as a crucial enzyme that serves as a procoagulant by activating platelets and producing fibrin and factors such as Va, VIIIa and XIII. One of the major SNPs implicated in the *PT* gene is rs1799963 as a G20210A variant, which contains an amino acid amendment between G and A at position 20210 and is associated with elevated PT circulation levels and the risk of venous thromboembolism [45]. MTHFR is considered the commonest studied variant in the human gene and is a folate-dependent enzyme that converts 5,10-methyl tetrahydrofolate to 5-methyl tetrahydrofolate, which is one of the important contributors in the regulation of plasma homocysteine levels [46]. Abnormal levels of homocysteine are associated with an increased risk of venous thrombosis. At amino acid position 222, a variant from C to T is substituted at nucleotide 677 from valine to alanine [47].

Until now, the exact role and its impact between thrombophilia, molecular and genetic variants in infertile women have not been documented accurately [48]. Hereditary thrombophilia is considered a risk factor in reproductive disorders [49]. However, in RPL [50], and then in cases of IVF [51], limited studies have been documented that have tried to connect infertility in women and genetic variants of thrombophilia [52,53]. A couple of studies have been documented in the Saudi population that were focused on recurrent miscarriage [54] that also included healthy subjects [55], and both of these studies showed positive associations. However, based on these studies, we attempted to develop a bridge between FI and thrombophilic variants, as these studies were restricted and did not determine whether these variants played any role in FI women. The objective of this case–control study was to explore the possible association between the G1691A/rs6020, G20210A/rs1799963 and C677T/rs1801133 variants of thrombophilia in the *FVL*, *PT(FII)* and *MTHFR* genes among Saudi infertile women.

## 2. Materials and Methods

### 2.1. Ethical Clearance for the Enrollment of Fertile and Infertile Women

This study was designed and implemented at G-141 Laboratory in the CLS Department at King Saud University (KSU). Ethical approval was received from the IRB committee, College of Medicine, at KSU. The concept of this study was explained to all participants prior to their involvement and after their agreement; finally, 200 Saudi women signed the informed consent form. This study protocol followed the Declaration of Helsinki. Of the Saudi women, 100 were infertile (cases) and 100 were healthy controls or fertile. All the women were enrolled from the outpatient clinic of the Department of Obstetrics and Gynecology within the hospital premises of KKUH. The selection of infertile and fertile women was based on inclusion and exclusion criteria. All the infertile women ($n = 100$) were regularly followed up for the treatment by visiting the outpatient clinic during their prescribed time of appointment. The inclusion criteria for infertile women were based on the confirmation of infertility from the clinician, not having been able to conceive for more than 1100 days and a family history of infertility; similar criteria were used to exclude control women. The exclusion criteria for FI cases were the absence of a family history of infertility and having been able to conceive a minimum of one or more children; similar criteria were used for the inclusion of control subjects, incorporating individuals with normal menstrual cycles. All the women ($n = 200$) involved in this study were between 18 and 44 years of age, with the case subjects being between 21 and 44 years of age and the control subjects being between 18 and 43 years of age. Clinical data were collected through a well-designed questionnaire, and 4 mL of venous blood was collected from each patient. Finally, 400 mL of serum and 400 mL of EDTA blood were collected from the 200 women. In this study, the sample size was calculated, resulting in the inclusion of 100 cases and 100 controls [56].

### 2.2. Anthropometric Measurements of the 200 Participants

In this study, we included height and weight to calculate body mass index (BMI) levels in all the recruited women based on the WHO's protocol [57]. The metric protocol was used to record weight in kilograms (kg) and height in centimeters (cm) to compute BMI. The BMI was categorized into underweight ($<18.5$ kg/m$^2$), original weight ($18.5$–$24.9$ kg/m$^2$), overweight ($25.0$–$29.9$ kg/m$^2$), Phase-I obesity ($30.0$–$34.9$ kg/m$^2$) and Phase-II-and-above obesities ($>35.0$ kg/m$^2$). In this study, we have recorded the ages for all the participants and family histories for infertile women. Expert women nurses measured BMI, and blood was also drawn during their visits to the outpatient clinic.

### 2.3. Serum Sample Experiments

In this study, 2 mL of coagulant blood was collected into a red tube containing silica particles, which function as a clot activator and coagulate the transferred venous blood. The collected blood in the vials were well mixed to coagulate the blood before centrifugation at 3500 rpm for 5 min to separate the serum on the top and the blood cells on the bottom. Next, the serum was transferred to 2 mL of Eppendorf tubes and aliquoted serum was used to measure hormones such as follicle-stimulating hormone (FSH), luteinizing hormone (LH) and thyroid-stimulating hormone (TSH). The remaining serum samples were stored at $-80$ °C for further analysis of the serum levels.

### 2.4. Amplification of the Thrombophilic Variants

Amplification analysis was performed by extracting 200 genomic DNAs from 200 peripheral blood samples collected in an EDTA tube. The protocol was instructed to perform the experiment using the Qiagen kit of the DNA extraction kit from the blood. All the DNA samples were labeled and stored at $-80$ °C after the completion of the experiment. The next day, a NanoDrop spectrophotometer was applied to measure the DNA quality. Finally, 200 DNA samples were converted into 20 ng/mL and further used for the amplification process. The protocol of genotyping analysis was carried out via polymerase chain reaction

(PCR) using the thermal cycler of Applied Biosystem for amplifying the G1691A, G20210A and C677T variants as initial denaturation was started with the help of 95 °C for 5 min and denaturation at 95 °C for 30 s. The annealing temperatures varied for each variant (displayed in Table 1), and extension and final extension were carried out 72 °C but varied for 45 s and 5 min. Qiagen master mix was used with a final reaction volume of 50 μL, and PCR amplification was conducted for 1.25–1.36 h. The reaction was set up for 35 cycles, and when the experiment was completed, the PCR was held at 4 °C. After confirmation of the presence of undigested PCR products for all three variants, precise restriction enzymes (Table 1) were used and digested with the PCR products at 37 °C for 2 h using NEB fastest restriction enzymes (HindIII and HinfI). Both the digested and undigested PCR products were run on 2.5% of ethidium-stained agarose gel. Analysis was carried out based on differences in band sizes. All the details of the SNPs, variants, primers, restriction enzymes and digested and undigested PCR products are shown in Table 1.

### 2.5. Statistical Analysis

The statistical analysis was completed using SPSS software (version 27.0). In this study, categorical and numerical variables were used. Numerical/categorical data are shown as mean ± standard deviation and total number of percentages. HWE analysis and genotype and allele frequencies were studied using SNPStat, while logistic regression models and clinical characteristics using *t*-tests were studied with SPSS software. ANOVA analysis was studied using Jamovi software Version 2.3 for the thrombophilic SNPs with Kruskal–Wallis tests, and generalized multifactorial dimensionality reduction (GMDR) analysis was implemented to study (i) gene–gene interactions, (ii) dendrogram analysis and (iii) depletion models in the study groups. The *p*-value was found to be significant if the value was <0.05 ($p < 0.05$).

**Table 1.** Involvement of thrombophilic SNPs and their details.

| Gene | rsnumber | Region | Location | Forward Primer | Reverse Primer | PCR Size | $T_m$ | Enzyme | RFLP Analysis |
|------|----------|--------|----------|----------------|----------------|----------|-------|--------|---------------|
| *FVL* | rs6020 | G1691A | Exon 10 | TCAGGCAGGAACAACACCAT | GGTTACTTCAAGGACAAAATACCT | 241 bp | 58 °C | HindIII | G-241bp; A-209/32 bp |
| *PT/FII* | rs1799963 | G20210A | 3′UTR | TCTAGAAACAGTTGCCTGGC | ATAGCACTGGGAGCATGAAGCAAG | 2345 bp | 60 °C | HindIII | G-345bp; A-322/23 bp |
| *MTHFR* | rs1801133 | C677T | Exon 4 | TGAAGGAGAAGGTGTGCTGA | AGGACGGTGCGGTGAGAGTG | 198 bp | 68 °C | HinfI | C-198bp: T-175/23 bp |

Tm = Melting temperature; PCR = Polymerase chain reaction; RFLP = Restriction fragment length polymorphism.

## 3. Results

### 3.1. Characteristics of the Women

　　In this case–control study, we enrolled 200 Saudi women, of which 100 were infertile and 100 fertile, aged between 18 and 44 years. The mean age ($30.79 \pm 5.36$ vs. $31.39 \pm 6.70$ years), height ($157.61 \pm 5.04$ vs. $159.02 \pm 6.88$ cm), weight ($73.88 \pm 11.27$ vs. $77.56 \pm 11.86$ kg) and BMI ($29.41 \pm 4.43$ kg/m$^2$ vs. $30.68 \pm 4.53$) were found to be high but non-significant ($p > 0.05$) in the controls. Additionally, the LH levels ($5.57 \pm 0.46$ vs. $6.95 \pm 2.36$ IU/mL; $p < 0.0001$) and LH/FSH ratio ($0.77 \pm 0.10$ vs.$1.37 \pm 0.86$; $p < 0.0001$) were also found to be high in the control women. Both the FSH ($7.33 \pm 0.77$ vs. $6.08 \pm 2.43$ IU/mL; $p = 0.001$) and TSH levels ($2.50 \pm 0.31$ vs. $2.16 \pm 0.31$ IU/mL; $p = 0.01$) were found to be high in the infertile women. In this study, all FI cases were found to have 100% infertility and none of them were pregnant. A family history of infertile women was found to be 53%, while the controls did not have any family history of infertility ($p < 0.0001$). All of the above results in the infertile and fertile women are specified in Table 2.

**Table 2.** Clinical details adopted from infertile and non-infertile Saudi women.

| Women's Data | Infertile Cases (*n* = 100) | Fertile Controls (*n* = 100) | *p*-Value |
|---|---|---|---|
| Age (years) | $30.79 \pm 5.36$ | $31.39 \pm 6.70$ | 0.485 |
| Height (cm) | $157.61 \pm 5.04$ | $159.02 \pm 6.88$ | 0.099 |
| Weight (kg) | $73.88 \pm 11.27$ | $77.56 \pm 11.86$ | 0.02 |
| BMI (kg/m$^2$) | $29.41 \pm 4.43$ | $30.68 \pm 4.53$ | 0.04 |
| FSH (IU/mL) | $7.33 \pm 0.77$ | $6.08 \pm 2.43$ | 0.001 |
| LH (IU/mL) | $5.57 \pm 0.46$ | $6.95 \pm 2.36$ | <0.0001 |
| LH/FSH ratio | $0.77 \pm 0.10$ | $1.37 \pm 0.86$ | <0.0001 |
| TSH (IU/mL) | $2.50 \pm 0.31$ | $2.16 \pm 0.31$ | 0.01 |
| Infertility | 100 (100%) | 00 (00%) | <0.0001 |
| Family history of FI | 53 (53%) | 00 (00%) | <0.0001 |

BMI = Body mass index; FSH = Follicle-stimulating hormone; LH = Luteinizing hormone; TSH-Thyroid-stimulating hormone.

### 3.2. HWE Analysis of the Thrombophilic Variants

　　Among the 13 thrombophilic genes, we have selected only three genes in this study, and each SNP was selected from the *FVL* (G1691A), *PT/FII* (G20210A) and *MTHFR* (C677T) genes. To inspect the impact of each selected SNP from the genes on FI, we performed HWE analysis in both the controls and cases. All details are explained in Table 3. In this study, all controls in the G1691A ($\chi^2 = 0.002$; $p = 0.95$), G2021A ($\chi^2 = 0.01$; $p = 0.91$) and C677T ($\chi^2 = 0.03$; $p = 0.84$) SNPs were found to be consistent ($p > 0.05$). Unfortunately, none of the infertile women were found to be under a significant association in the HWE analysis ($p < 0.05$) for the G1691A ($\chi^2 = 9.77$; $p = 0.001$), G2021A ($\chi^2 = 4.78$; $p = 0.02$) or C677T ($\chi^2 = 3.93$; $p = 0.04$) SNPs.

**Table 3.** HWE analysis between the SNPs of infertile and fertile women.

| | Fertile Women (*n* = 100) | | | Infertile Women (*n* = 100) | | |
|---|---|---|---|---|---|---|
| | G1691A | G20210A | C677T | G1691A | G20210A | C677T |
| HWE analysis | 0.01 | 0.01 | 0.15 | 0.03 | 0.04 | 0.16 |
| $\chi^2$ | 0.002 | 0.01 | 0.03 | 9.77 | 4.78 | 3.93 |
| *p*-Value | 0.95 | 0.91 | 0.84 | 0.001 | 0.02 | 0.04 |

### 3.3. Genotyping Analysis of the Thrombophilic Variants in the Infertile and Fertile Women

Table 4 shows the genotypes present in the G1691A, G20210A and C677T variants in the *FVL*, *FII* and *MTHFR* genes in both the cases and controls. The GG, GA and AA genotypes of the *FVL* gene present in the cases and controls appeared to be 95%, 4%, 1% and 99%, 1%. Genotyping analysis was conducted for the cases and controls and were as follows: GA vs. GG (OR-4.168 (95%CI: 0.45–37.97); $p$ = 0.171) and AA vs. GG (OR-2.084 (95%CI: 0.06–62.84); $p$ = 0.665). The different genetic models included GA + AA vs. GG (OR-5.211 (95%CI: 0.59–45.42); $p$ = 0.097), AA + GA vs. GG (OR-0.242 (95%CI: 0.02–2.21); $p$ = 0.174) and GG + GA vs. AA (OR-0.495 (95%CI: 0.016–14.92); $p$ = 0.679). Meanwhile, the G20210A SNP present in the FI cases was found to have 93% GG, 6% GA and 1% AA genotypes, while in the control subjects, it was found to have 98% GG and 2% GA genotypes. The statistical analysis was as follows: GA vs. GG (OR-3.161 (95%CI: 0.62–16.06); $p$ = 0.144) and AA vs. GG (OR-2.108 (95%CI: 0.06–63.55); $p$ = 0.661). The various genetic models included GA + AA vs. GG (OR-3.688 (95%CI: 0.74–18.21); $p$ = 0.088), AA + GA vs. GG (OR-0.319 (95%CI: 0.06–1.62); $p$ = 0.149) and GG + GA vs. AA (OR-0.495 (95%CI: 0.01–14.92); $p$ = 0.679). However, the AA genotype was completely absent in the control population for the G1691A and G20210A variants present in the *FVL* and *FII* genes. Finally, the CC, CT and TT genotypes present in the infertile women were found to be 74%, 21%, 5% and 72%, 26%, 2% in the controls for the C677T variant in *MTHFR* gene. Genetic analysis showed CT vs. CC (OR-0.785 (95%CI: 0.40–1.52); $p$ = 0.473) and TT vs. CC (OR-2.432 (95%CI: 0.45–12.94); $p$ = 0.284). The three genetic models included CT + TT vs. CC (OR-0.903 (95%CI: 0.48–1.68); $p$ = 0.751), TT + CC vs. CT (OR-1.322 (95%CI: 0.68–2.54); $p$ = 0.404) and CC + CT vs. TT (OR-0.387 (95%CI: 0.07–2.04); $p$ = 0.249).

**Table 4.** Genotype frequencies of the thrombophilic variants in infertile and fertile women.

| Gene (rsnumber) | Genotypes | Infertile (*n* = 100) | Fertile (*n* = 100) | OR (95%CI) | *p*-Value |
|---|---|---|---|---|---|
| *FVL* (rs6020) | GG | 95 (95%) | 99 (99%) | 1[Reference] | 1[Reference] |
| | GA | 04 (04%) | 01 (01%) | 4.168 (0.45–37.97) | 0.171 |
| | AA | 01 (01%) | 00 (00) | 2.084 (0.06–62.84) | 0.665 * |
| | GA + AA vs. GG | 05 (05%) | 01 (01%) | 5.211 (0.59–45.42) | 0.097 |
| | AA + GG vs. GA | 96 (96%) | 99 (99%) | 0.242 (0.02–2.21) | 0.174 |
| | GG + GA vs. AA | 99 (99%) | 100 (100%) | 0.495 (0.016–14.92) | 0.679 * |
| *PT/FII* (rs20210) | GG | 93 (93%) | 98 (98%) | 1[Reference] | 1[Reference] |
| | GA | 06 (06%) | 02 (02%) | 3.161 (0.62–16.06) | 0.144 |
| | AA | 01 (01%) | 00 (00) | 2.108 (0.06–63.55) | 0.661 * |
| | GA + AA vs. GG | 07 (07%) | 02 (02%) | 3.688 (0.74–18.21) | 0.088 |
| | AA + GG vs. GA | 94 (94%) | 98 (98%) | 0.319 (0.06–1.62) | 0.149 |
| | GG + GA vs. AA | 99 (99%) | 100 (100%) | 0.495 (0.01–14.92) | 0.679 * |
| *MTHFR* (rs1801133) | CC | 74 (74%) | 72 (72%) | 1[Reference] | 1[Reference] |
| | CT | 21 (21%) | 26 (26%) | 0.785 (0.40–1.52) | 0.473 |
| | TT | 05 (05%) | 02 (02%) | 2.432 (0.45–12.94) | 0.284 |
| | CT + TT vs. CC | 26 (26%) | 28 (28%) | 0.903 (0.48–1.68) | 0.751 |
| | TT + CC vs. CT | 79 (79%) | 74 (74%) | 1.322 (0.68–2.54) | 0.404 |
| | CC + CT vs. TT | 95 (95%) | 98 (98%) | 0.387 (0.07–2.04) | 0.249 |

* Yates correction between the genotypes.

### 3.4. Allele Frequency Studies in the Cases and Controls of the Thrombophilic Variants

The allele frequencies present in the G1691A, G20210A and C677T SNPs are shown in Table 5. The level of risk alleles (A allele) found in G16191A was 3% and 0.5% in both the FI

cases and controls, 4% and 1% in the cases and controls of the G20210A SNP and 15.5% and 15% in the T alleles of the C677T SNP. However, the normal allele (G allele) level was found to be 97% and 99.5% in the G6191A SNP, 96% and 99% in the G20210A SNP and, finally, 84.5% and 85% in the C677T SNP. The statistical analysis conducted between the FI cases and controls and analysis showed results as follows: G1691A (OR-6.155 (95%CI: 0.73–51.59); $p = 0.056$), G20210A (OR-4.125 (95%CI: 0.86–19.67); $p = 0.054$) and C677T (OR-1.039 (95%CI: 0.60–1.79); $p = 0.889$).

**Table 5.** Allele frequencies in the G1691A, G20210A and C677T variants in infertile and fertile women.

| Gene(s) (rsnumber) | Alleles | Infertile (*n* = 100) | Fertile (*n* = 100) | OR (95%CI) | *p*-Value |
|---|---|---|---|---|---|
| *FVL* (rs6020) | G | 194 (97%) | 199 (99.5%) | 1[Reference] | 1[Reference] |
|  | A | 06 (03%) | 01 (0.5%) | 6.155 (0.73–51.59) | 0.056 |
| *PT/FII* (rs20210) | G | 192 (96%) | 198 (99%) | 1[Reference] | 1[Reference] |
|  | A | 08 (04%) | 02 (01%) | 4.125 (0.86–19.67) | 0.054 |
| *MTHFR* (rs1801133) | C | 169 (84.5%) | 170 (85%) | 1[Reference] | 1[Reference] |
|  | T | 31 (15.5%) | 30 (15%) | 1.039 (0.60–1.79) | 0.889 |

*3.5. Logistic Regression and ANOVA Analysis Studied among the Infertile Women*

Table 6 shows calculations of the regression model, conducted between the thrombophilic genotypes involved in this study and the involvement of seven covariates: (i) Age, (ii) weight, (iii) BMI, (iv) FSH, (v) LH, (vi) TSH and (vii) LH/FSH ratio. None of the covariates were found to be associated ($p > 0.05$). Table 7 presents the ANOVA analysis results between the three genotypes in each SNP among the specified covariates. An overall elevated value of the covariates among those aged $31.09 \pm 5.27$ years was found in the CC genotype of the C677T SNP; weight was found to be $81.00 \pm 0.00$ kg in the AA genotype of the G20210A SNP; BMI was found to be $34.20 \pm 0.00$ kg/m$^2$ among the AA genotype in the G1691A SNP. An elevated FSH value of $7.65 \pm 0.51$ IU/mL was found in the GA genotype of the G1691A SNP. Elevated LH ($6.04 \pm 0.81$ IU/mL) and LH/FSH ratio ($0.84 \pm 0.15$) values were found in the TT genotype of the C677T SNP. Finally, an elevated TSH level of $2.94 \pm 0.00$ IU/mL was found in the AA genotype of the G20210A SNP. These results highlight that none of the regression and ANOVA analyses were associated with the thrombophilic variants and covariates ($p > 0.05$).

**Table 6.** Multinomial logistic regression analysis of the thrombophilic variants in infertile women.

| Covariates | R-Value | Adjusted R-Square Value | Standardized β-Coefficient for rs6020 | Standardized β-Coefficient for rs20210 | Standardized β-Coefficient for rs1801133 | F | *p*-Value |
|---|---|---|---|---|---|---|---|
| Age (years) | 0.211 | 0.015 | −0.177 | −0.052 | −0.109 | 1.495 | 0.221 |
| Weight (kg) | 0.153 | −0.007 | −0.130 | 0.014 | −0.077 | 0.765 | 0.516 |
| BMI (kg/m$^2$) | 0.089 | −0.023 | −0.012 | −0.001 | −0.088 | 0.254 | 0.858 |
| FSH (IU/mL) | 0.135 | −0.013 | 0.065 | −0.048 | −0.108 | 0.593 | 0.621 |
| LH (IU/mL) | 0.165 | −0.003 | −0.033 | −0.076 | 0.142 | 0.897 | 0.446 |
| LH/FSH ratio | 0.199 | 0.010 | −0.077 | −0.022 | 0.183 | 1.324 | 0.271 |
| TSH (IU/mL) | 0.137 | −0.012 | 0.010 | 0.028 | 0.134 | 0.611 | 0.610 |

BMI = Body mass index; FSH = Follicle-stimulating hormone; LH = Luteinizing hormone; TSH = Thyroid-stimulating hormone.

**Table 7.** ANOVA analysis between the thrombophilic genotypes in infertile women.

| Covariates | rs6020 | | | | rs20210 | | | | rs1801133 | | | |
|---|---|---|---|---|---|---|---|---|---|---|---|---|
| | GG = 95 | GA = 04 | AA = 01 | *p*-Value | GG = 93 | GA = 06 | AA = 01 | *p*-Value | CC = 74 | CT = 21 | TT = 05 | *p*-Value |
| Age (years) | $31.00 \pm 5.40$ | $27.25 \pm 2.06$ | $25.00 \pm 0.00$ | 0.218 | $30.85 \pm 5.37$ | $30.00 \pm 6.03$ | $30.00 \pm 0.0$ | 0.922 | $31.09 \pm 5.27$ | $30.19 \pm 5.56$ | $28.80 \pm 6.42$ | 0.557 |
| Weight (kg) | $74.32 \pm 10.87$ | $63.15 \pm 18.22$ | $75.00 \pm 0.00$ | 0.151 | $73.87 \pm 11.18$ | $72.85 \pm 14.37$ | $81.00 \pm 0.00$ | 0.802 | $74.69 \pm 10.90$ | $70.52 \pm 12.68$ | $76.04 \pm 9.88$ | 0.299 |
| BMI (kg/m$^2$) | $29.47 \pm 4.23$ | $26.63 \pm 8.27$ | $34.20 \pm 0.00$ | 0.253 | $29.42 \pm 4.44$ | $28.95 \pm 4.98$ | $30.90 \pm 0.00$ | 0.916 | $29.73 \pm 4.26$ | $28.15 \pm 4.99$ | $29.88 \pm 4.44$ | 0.346 |
| FSH (IU/mL) | $7.32 \pm 0.78$ | $7.65 \pm 0.51$ | $7.40 \pm 0.00$ | 0.703 | $7.34 \pm 0.79$ | $7.32 \pm 0.43$ | $6.80 \pm 0.00$ | 0.240 | $7.39 \pm 0.78$ | $7.17 \pm 0.75$ | $7.22 \pm 0.74$ | 0.488 |
| LH (IU/mL) | $5.58 \pm 0.46$ | $5.33 \pm 0.49$ | $5.90 \pm 0.00$ | 0.444 | $5.59 \pm 0.47$ | $5.35 \pm 0.24$ | $5.70 \pm 0.00$ | 0.800 | $5.56 \pm 0.46$ | $5.53 \pm 0.28$ | $6.04 \pm 0.81$ | 0.064 |
| LH/FSH ratio | $0.77 \pm 0.10$ | $0.70 \pm 0.10$ | $0.80 \pm 0.00$ | 0.373 | $0.77 \pm 0.10$ | $0.73 \pm 0.06$ | $0.84 \pm 0.00$ | 0.482 | $0.76 \pm 0.10$ | $0.78 \pm 0.09$ | $0.84 \pm 0.15$ | 0.194 |
| TSH (IU/mL) | $2.50 \pm 0.31$ | $2.45 \pm 0.34$ | $2.63 \pm 0.00$ | 0.870 | $2.50 \pm 0.31$ | $2.39 \pm 0.22$ | $2.94 \pm 0.00$ | 0.246 | $2.47 \pm 0.30$ | $2.55 \pm 0.35$ | $2.61 \pm 0.31$ | 0.407 |

BMI, body mass index; FSH, follicle-stimulating hormone; LH, leutinizing hormone; TSH, thyroid-stimulating hormone.

*3.6. Combined Genotyping and Allele Frequencies among the Thrombophilic Variants in the Case and Control Women*

In this study, we also created Table 8, which consists of the combined genotypes and allele frequencies for the G1691A, G20210A and C677T SNPs present in all the cases compared to all the controls to explore the potential cumulative influence of thrombophilic genotypes. The combination for the normal genotype was calculated as GG/GG/CC (87.33% vs. 89.67%), while the heterozygous genotype was shown to be GA/GA/CT (10.33% vs. 9.67%) and the homozygous variant was indicated as AA/AA/TT (2.23% vs. 0.67%). The normal and variant alleles were confirmed as G/G/C (92.5% vs. 94.5%) and A/A/T (7.5% vs. 5.5%). The overall genetic analysis for the heterozygous (OR-1.098 (95%CI: 0.64–1.87); $p = 0.732$) and homozygous variant genotypes (OR-3.594 (95%CI: 0.73–17.46); $p = 0.090$) was found to be inconsistent. The allele frequencies were also found to have non-significant associations (OR-1.393 (95%CI: 0.87–2.21); $p = 0.160$). Additionally, the dominant (12.67% vs. 10.33%; OR-1.259 (95%CI: 0.76–2.08); $p = 0.374$) and co-dominant (89.67% vs. 75%; OR-1.077 (95%CI: 0.63–1.83); $p = 0.785$) models showed negative associations, while, finally, the recessive model (97.67% vs. 91.33%; OR-3.972 (95%CI: 1.69–9.29); $p = 0.0006$) showed positive associations.

**Table 8.** Combined genotype and allele frequencies among the thrombophilic variants present in infertile and fertile women.

| Genotypes | Infertile ($n = 100$) | Fertile ($n = 100$) | OR (95%CI) | $p$-Value |
|---|---|---|---|---|
| GG/GG/CC genotypes | 262 (87.33%) | 269 (89.67%) | 1 Reference | 1 Reference |
| GA/GA/CT genotypes | 31 (10.33%) | 29 (9.67%) | OR-1.098 (95%CI: 0.64–1.87) | 0.732 |
| AA/AA/TT genotypes | 07 (2.33%) | 02 (0.67%) | OR-3.594 (95%CI: 0.73–17.46) | 0.090 |
| GA/GA/CT + AA/AA/TT vs. GG/GG/CC | 38 (12.67%) | 31 (10.33%) | OR-1.259 (95%CI: 0.76–2.08) | 0.374 |
| AA/AA/TT + GG/GG/CC vs. GA/GA/CT | 269 (89.67%) | 225 (75%) | OR-1.077 (95%CI: 0.63–1.83) | 0.785 |
| GG/GG/CC + GA/GA/CT vs. AA/AA/TT | 293 (97.67%) | 274 (91.33%) | OR-3.972 (95%CI: 1.69–9.29) | 0.0006 * |
| G/G/C alleles | 555 (92.5%) | 567 (94.5%) | 1 Reference | 1 Reference |
| A/A/T alleles | 45 (7.5%) | 33 (5.5%) | OR-1.393 (95%CI: 0.87–2.21) | 0.160 |

* indicates statistical significant which is $p < 0.05$.

*3.7. Predicted Studies of GMDR Analysis in the Case and Control Women*

In this study, we included all the genotype data from the cases and controls and tabulated in Table 9 to show the gene–gene interaction analysis. All three thrombophilic SNPs were combined to study the gene–gene interactions, dendrograms (Figure 1) and depletion models (Figure 2). The combination of G20210A/C677T SNPs (OR-6.126 (95%CI: 1.31–28.49); $p = 0.0096$) and G20210A/G1691A/C677T SNPs (OR-6.270 (95%CI: 1.75–22.36); $p = 0.0016$) showed positive associations, while the G20210A SNP showed negative associations (OR-1.454 (95%CI: 0.72–2.91); $p = 0.2903$). The dendogram analysis revealed moderate association with G20210A SNP and redundancy associations between the G1691A and C677T SNPs. High, moderate and low risks were used to analyze the overall findings of this study. High risks are defined by dark boxes, lower risks by light boxes and the lack of data by blank or white boxes. The overall analysis confirmed that moderate risk exists between the thrombophilic SNPs, indicating combinations of high, low and no risk are involved in the studied population.

**Table 9.** Estimation of gene–gene interactions among the thrombophilic variants in infertile women.

| Model No. | Best Combinations | Training Accuracy | Testing Accuracy | CVC | Training Sensitivity | Training Specificity | $\chi^2$ | OR (95%CI) | *p*-Value | F-Measure | Kappa |
|---|---|---|---|---|---|---|---|---|---|---|---|
| 1 | G20210A | 0.5278 | 0.495 | 6/10 | 0.8 | 0.2668 | 1.118 | 1.454 (95%CI: 0.72–2.91) | 0.2903 | 0.6316 | 0.0667 |
| 2 | G20210A + C677T | 0.5511 | 0.515 | 7/10 | 0.1222 | 0.9778 | 6.715 | 6.126 (95%CI: 1.31–28.49) | 0.0096 | 0.2136 | 0.1 |
| 3 | G20210A + G1691A + C677T | 0.5700 | 0.565 | 10/10 | 0.1778 | 0.9667 | 9.944 | 6.270 (95%CI: 1.75–22.36) | 0.0016 | 0.2936 | 0.1444 |

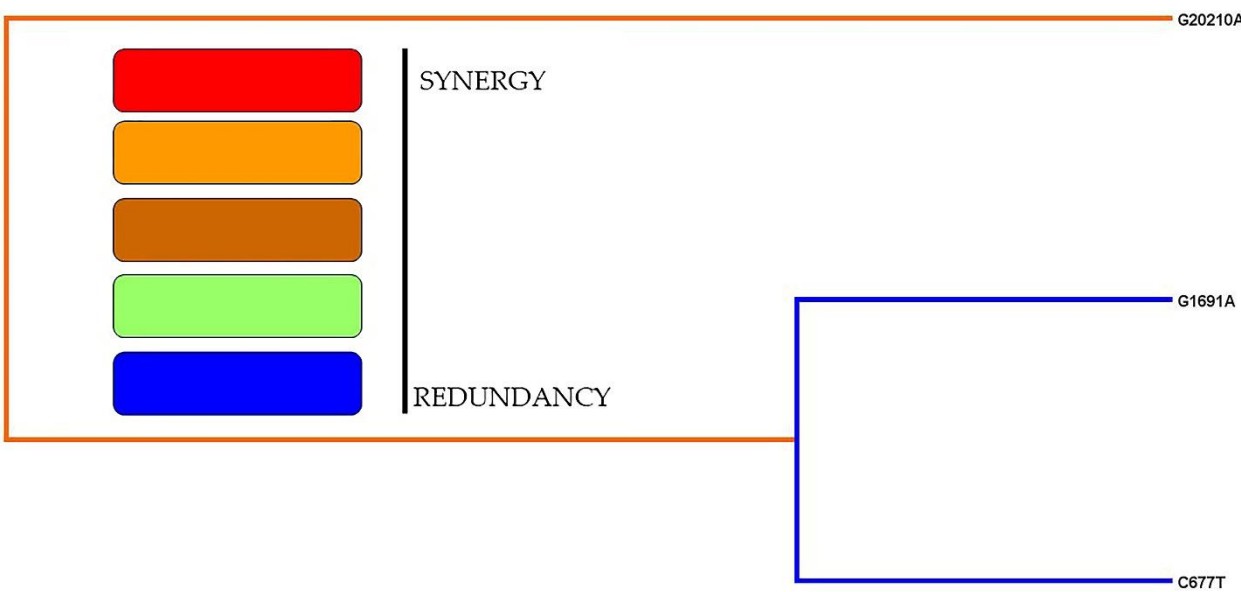

**Figure 1.** Dendogram analysis of the thrombophilic variants.

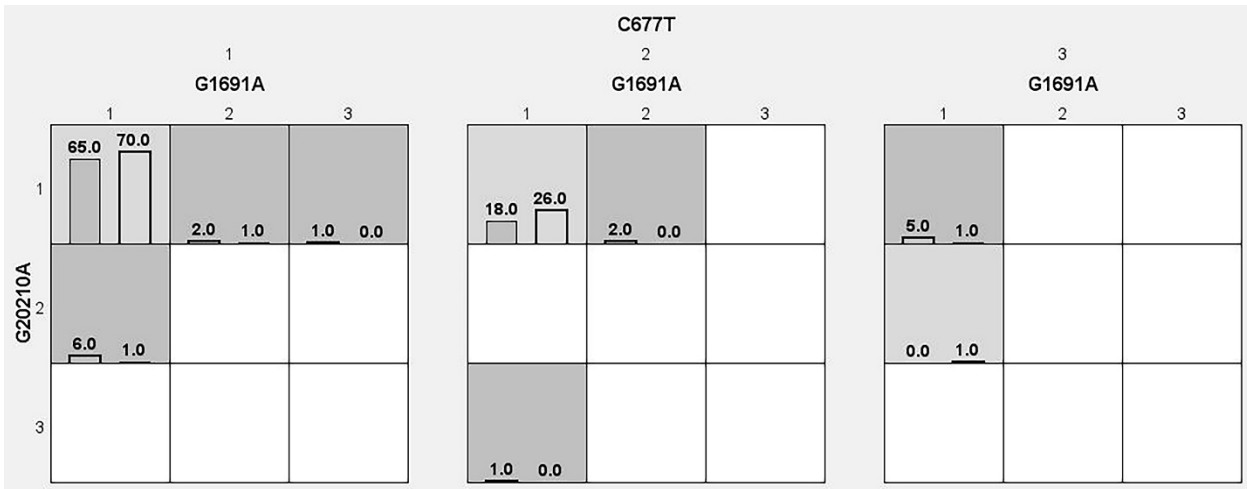

**Figure 2.** Analysis of the graphical depletion model and its predicted risks in the thrombophilic variants.

## 4. Discussion

In this study, the thrombophilic variants were studied in rs6020 (G1691A), rs1799963 (G20210A) and rs1801133 (C677T) SNPs of the GG/CC, GA/CT and AA/TT genotypes. To maximize power to detect the accuracy of the heterozygous variants present in the thrombophilic variants, we opted to include Saudi women in this case–control study, which can be considered as one of the strengths of this study. We recruited 100 infertile women and 100 fertile women based on confirmation from gynecologists and the patients' history. Thrombophilia itself is defined as a predisposition toward thrombosis, either hereditary or diagnosed postnatally [48,58]. Thrombophilia also includes rare inherited defects that lead to the enhancement of coagulation [21]. Mutations, variants or SNPs in thrombophilia may disrupt the bodies coagulation mechanism, resulting in an increased susceptibility to producing blood clots improperly. This may have serious consequences for human health, since aberrant blood clot development can lead to disorders such as deep vein thrombosis, pulmonary embolism, stroke and other thrombotic events. Furthermore, it may have an impact on the female reproductive system since FI is a human-effecting health system. Based on these factors, this study was designed to investigate the role of thrombophilic

SNPs in Saudi infertile women. The current study results showed a negative association in the genotype and allele frequencies. Yates correction was applied towards the absence of AA genotypes in the G1691A and G20210A SNPs for conservative adjustments when dealing with a small sample size. The 100 FI cases and 100 controls recruited to this study can be considered a small sample, which is also one of the limitations of this study. For small sample sizes, Yates adjustment should be used with caution, since it may result in a loss of statistical power, making it more difficult to discover genuine relationships when they exist. In addition to this, logistic regression models and ANOVA analysis were employed. However, a positive association was found in the recessive models of the combined genotype frequencies among the thrombophilic SNPs ($p = 0.0006$). A moderate association was found in the GMDR model using gene–gene interaction, dendrograms and depletion models.

The absence or disappearance of homozygous variants in a specific population-based case–control study could have several potential indications based on the context of the study, as well as other factors, such as a rare variant. In this case, the AA genotypes of both the G1691A and G20210A SNPs can be considered rare variants among the several human diseases in Saudi Arabia, especially as it was totally disappearing in the control population [59–61]. Other factors such as a low penetrance level, loss of functional variants, negative selection, genetic drift and considering the low sample size necessitate conducting further research. However, it is considered critical to evaluate these results in the context of the particular research issue and the variants' underlying biology. In certain circumstances, the lack of homozygous individuals may be anticipated and instructive, but in others, additional examination may be required to understand the consequences for the condition or trait for this study. Finally, if any of the control individuals develop homozygous variants in the Saudi population, then it can be considered as a disease-causing variant in the future within their own family, as well as in the Saudi community.

Inherited thrombophilia variants/mutations are considered a leading cause of recurrent spontaneous miscarriage, infertility, RPL, implantation failure, issues during IVF and ovarian hyperstimulation syndrome. Thrombophilic variants disrupt trophoblast differentiation, as well as placental vascularization, resulting in fetal growth restrictions, pregnancy failures, infertility, placental insufficiencies and, eventually, miscarriages [62–64]. The role of inherited thrombophilia does not show a genetic association with Saudi women regarding infertility, supported by previous studies among unexplained infertility in female subjects [52,53,59,61,65], as well as by a large prospective study in the Arab community [66]. One of the possible explanations for the contradictory results of the previously mentioned studies might be the influence of epigenetic and environmental factors in the aforementioned studied groups. However, data on the discovered FI variants and their prognostic and therapeutic implications are missing [67]. A recently published systematic review and meta-analysis study on thrombophilic gene polymorphisms and RPL suggests that they may be used as effective clinical indicators to assess RPL risks or to assist unexplained RPL women uncover plausible factors, as well as to allow for focused treatment [50]. One of the published meta-analyses on 99 genetic analysis studies of the C677T variant in the *MTHFR* gene in venous thrombosis confirmed the contribution toward the development of pulmonary embolism [68].

One of the risk factors for FI is known to be advanced age, which is also responsible for pregnancy loss and other obstetric complications [69]. Women who conceive after 35 years of age are considered to be of advanced maternal age, associated with many gynecological complications and reproductive issues. Advanced maternal age also causes declination in ovarian reserve and oocyte competence. Additionally, it has also a strong impact on decreased fertility and increased infertility during a woman's life [70], such as a decline in ovarian reserve, a long time to conceive, a low rate of ovarian stimulation, unhealthy menstruation, increased risk of miscarriage, hormonal changes, high risk of multiple gestations/pregnancy complications, low pregnancy rate and high chances of having birth defects. In Saudi Arabia, there are many studies that support the above

complications in FI women [71–73]. In this study, we did not record the complications present in FI women, which can be considered another limitation of this study. In our study, 48% of the FI women were in the age range of 21–30 years, while 49% of them were between 31 and 40 years of age and 3% were above 40 years of age (i.e., 41–44 years). Regarding the control population, 7% of the women were found to be between 18 and 20 years of age, 37% between 21 and 30 years of age, 50% between 31 and 40 years of age and 6% between 41 and 43 years of age. Age was recorded during the enrollment of this study. On average, almost 50% of both the case–control populations were found to be between 31 and 40 years of age. In this study, the mean of the controls ($31.39 \pm 6.70$ years) was found to be high when compared to the FI cases ($30.79 \pm 5.36$ years). Simultaneously, this affected the anthropometric measurements such as weight ($77.56 \pm 11.86$ vs. $73.88 \pm 11.27$ kg) and BMI levels ($30.68 \pm 4.53$ vs. $29.41 \pm 4.43$ kg/m$^2$), which were found to be elevated in the control population. The control women were found to be obese, while FI cases were falling under the category of overweight ($25.1$–$29.9$ kg/m$^2$).

The SNPs present in the 3′UTR region on a gene play an important regulatory role in gene expression and post-transcriptional control. Furthermore, they can modulate silencing present on miRNA-dependent genes. The SNPs present in the 3′UTR region can influence the stability of mRNA, and this 3′UTR is known to be the non-coding region of mRNA that exist downstream of exon sequences. One of the advantages of studying the 3′UTR region is to determine the translation rate of mRNA into proteins, which further influences the binding of regulatory factors, controlling the initiation or elongation of translation and further modulating gene expression [74–77]. In our study, the G20210A SNP was present in the 3′UTR region, and 7% of FI women were found to have GA/AA genotypes, while a couple of control women had developed the GA genotype. Furthermore, the SNPs present in exons can modulate gene transcription and translation, and the SNPs present in introns can attack RNA splicing, genomic imprinting and lncRNAs, while the SNPs present at 5′UTR can promote translation [77]. In this study, the G1691A and C677T SNPs were present in Exon-10 and Exon-4 (Table 1); 5% of the SNPs were the GA and AA genotypes in the FI cases, and a single GA genotype was present in the control subjects; while in the C677T SNP, 26% of the CT and TT genotypes existed in the FI cases, and 28% of the CT and TT genotypes appeared in the control women. The CT and TT genotype frequencies were found to be high in the controls in comparison to the cases. A previous study in Saudi Arabia investigated the relationship between the C677T SNP and FI, and the prevalence of the CC, CT and TT genotypes was found to be 70.7%, 23.3% and 6% in the cases and 82%, 15.3% and 2.7% in the control women. We studied the C677T SNP as a trio combination for the thrombophilic variants, showing consistency with previous studies [78].

We found negative associations in all of the studied SNPs, and this was supported by previous studies of thrombophilia in the global population [41,79–82]. Overall, the negative association results indicate that there is no risk of the studied SNPs in human diseases [83,84]. In general, a relationship was already established between thermophobia and obesity through endogenous factors [85]. In our study, thrombophilic SNPs did not affect women with obesity in either the FI cases or the control population. However, in a recent case–report study, a 48-year-old Saudi obese patient was diagnosed with a heterogenous mutation in the G20210A SNP [86]. Additionally, this patient was diagnosed with diabetes and dyslipidemia. In our study, none of the women were diagnosed with diabetes or dyslipidemia. The FI cases involved in this study were confirmed only with obesity and infertility.

In this study, we selected a single SNP from a single gene for the 13 thrombophilic genes and we did not perform validation for our results, which can be considered the final limitations of this study. One of the advantages of this study is that we have recruited all the Saudi women and a clinician confirmed FI based on regular check-up visits.

## 5. Conclusions

The G1691A, G20210A and C677T SNPs play no role in Saudi infertile women, although this may be due to the small sample size. Future studies should be performed using a large sample and involve additional thrombophilic SNPs in the Saudi population.

**Author Contributions:** Conceptualization, I.A.K., A.A.A. and M.M.A.-H.; methodology, I.A.K., M.A.A., S.M.N. and M.M.A.-H.; software, M.A.A. and S.M.N.; validation, I.A.K. and A.A.A.; formal analysis, A.A.A. and M.A.A.; investigation, I.A.K., S.M.N. and M.M.A.-H.; resources, M.A.A., S.M.N. and M.M.A.-H.; data curation, I.A.K. and A.A.A.; writing—original draft preparation, I.A.K. and A.A.A.; writing—review and editing, I.A.K.; visualization, I.A.K., A.A.A. and M.M.A.-H.; supervision, I.A.K., A.A.A. and M.M.A.-H.; project administration, I.A.K.; funding acquisition, I.A.K. All authors have read and agreed to the published version of the manuscript.

**Funding:** This research was funded by Research supporting project number (RSPD2023R735), King Saud University, Riyadh, Saudi Arabia.

**Institutional Review Board Statement:** This study was conducted in accordance with the Declaration of Helsinki and approved by the Institutional Review Board (or Ethics Committee) of King Saud University (E-23–7917 and 11 July 2023) for studies involving humans.

**Informed Consent Statement:** Informed consent was obtained from all subjects involved in this study.

**Data Availability Statement:** All data are available within this study.

**Acknowledgments:** The authors would like to extend their sincere appreciation to the research supporting project number (RSPD2023R735), King Saud University, Riyadh, Saudi Arabia, for funding this project.

**Conflicts of Interest:** The authors declare no conflict of interest.

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
