# Peer review of "Molecular Screening of the Thrombophilic Variants Performed at G-141 Laboratory among Saudi Infertile Women"

_applsci, doi:10.3390/app132413028_

Round 1

Reviewer 1 Report

Comments and Suggestions for Authors

The manuscript required major correction

1.       There are a lot of errors in the manuscript that need to be corrected. The grammar needs to be improved.

2.       Abstract: The abstract should be clear and concise, stating the method used, findings, and conclusion.

3.       Line 38 - 43: Are they relevant? Pl, remove.

4.       There are a lot of inconsistencies in the manuscript for example in the abstract the population size mentioned was 200. However, in the methodology section 210 and 220 population sizes were mentioned.

5.       The rs numbers mentioned should be uniform throughout the manuscript. The rs numbers mentioned in the discussion section were different from those reported in the tables.

6.       No consistency in the unit format: The Format of writing units should be uniform all through

7.       The discussion needs to be improved.

Comments on the Quality of English Language

There are a lot of grammatical errors that need to be corrected. Therefore, the English language needs to be improved.

Author Response

Reviewer comments 1

The manuscript required major correction

  1. There are a lot of errors in the manuscript that need to be corrected. The grammar needs to be improved.
    Ans) The manuscript was edited the native experts and rectified the grammatical errors.

  2. Abstract: The abstract should be clear and concise, stating the method used, findings, and conclusion.

    Ans) We have edited the manuscript and updated the abstract as per your suggestion.

  3. Line 38 - 43: Are they relevant? Pl, remove.
    Ans) We have removed it.

  4. There are a lot of inconsistencies in the manuscript for example in the abstract the population size mentioned was 200. However, in the methodology section 210 and 220 population sizes were mentioned.
    Ans) We have rectified the error. In this study, we have used 100 cases and 100 controls.

  5. The rs numbers mentioned should be uniform throughout the manuscript. The rs numbers mentioned in the discussion section were different from those reported in the tables.
    Ans) Now, we have updated in the revised manuscript.

  6. No consistency in the unit format: The Format of writing units should be uniform all through
    Ans) The manuscript was edited with the native experts and revised and updated in the revised manuscript.

  7. The discussion needs to be improved.
    Ans) We have done our best in developing the discussion.

Reviewer 2 Report

Comments and Suggestions for Authors

In the article entitled "Molecular screening of thrombophilic variants among Saudi infertile women", Alageel, A.A. et al. aims to explore the association between G1691A, G20210A, and C677T variants of thrombophilia in FVL, FII and MTHFR genes among Saudi infertile women. The reviewer has the following comments:

1. Although the study holds some merit, in my opinion it is short of being a significant contribution to the field. The rationale behind some of the analysis are not clearly explained. The narrative needs to be significantly improved.
2. Based on what did the authors choose the three genes in the study among a total of 13 thrombophilic genes?  Also, did the choice of the genes influence the final conclusion that thrombophilic SNPs have no role and involvement in infertility?

3. For some of the result sections, no clear rationale or conclusions is provided summarizing the findings. A summary statement would greatly enhance readability and clarity.

Comments on the Quality of English Language

The quality of English language must be significantly improved to enhance the clarity of the article. Sometimes it is difficult to comprehend the meaning of the sentences. 
The article contains numerous typos and grammatical errors which must be corrected. 

Author Response

Reviewer comments 2

Comments and Suggestions for Authors

In the article entitled "Molecular screening of thrombophilic variants among Saudi infertile women", Alageel, A.A. et al. aims to explore the association between G1691A, G20210A, and C677T variants of thrombophilia in FVL, FII and MTHFR genes among Saudi infertile women. The reviewer has the following comments:

  1. Although the study holds some merit, in my opinion it is short of being a significant contribution to the field. The rationale behind some of the analyses are not clearly explained. The narrative needs to be significantly improved.
    Ans) Dear Reviewer, thank you for your valuable comments. The manuscript was edited with the native experts and updated in the revised manuscript.
  2. Based on what did the authors choose the three genes in the study among a total of 13 thrombophilic genes?  Also, did the choice of the genes influence the final conclusion that thrombophilic SNPs have no role and involvement in infertility?
    Ans) Among 13 thrombophilic genes, we have opted only 3 genes. The selection was based on its role in the Saudi population and we have confirmed selection of single SNP from 1 gene and selection of only 3 genes was one of the limitations of our study.

  3. For some of the result sections, no clear rationale or conclusions is provided summarizing the findings. A summary statement would greatly enhance readability and clarity.
    Ans) Now, the manuscript was edited with the native experts and updated in the revised manuscript. Now, we assume that we have justified the raised query.

Comments on the Quality of English Language

The quality of English language must be significantly improved to enhance the clarity of the article. Sometimes it is difficult to comprehend the meaning of the sentences. The article contains numerous typos and grammatical errors which must be corrected. 

The manuscript was edited with the native experts and both typos and grammatical errors were rectified.

Reviewer 3 Report

Comments and Suggestions for Authors

Abstract

Line 29:

 among G1691A, G20210A and C677T SNPs among thrombophilia – please rephrase to remove one among.

Line 38 to 43: I think it should be removed.

Line 56 to 59: not clear – please clarify the meaning.

Line 59: One of the common causes of infertility in both the genders or couples could be dysfunction of the reproductive system [7] – repetitive.

Line 62: another formulation for the percentage.

Please bundle all related information together and do not just repeat in different voice.

Line 69 to 84: I did not get the relation of obesity to the studied polymorphism. Please be concise and to the point.

Line 83 and 84: repetition.

Line 116-124: What is the relation of study to obesity???

Line 142: MTHFR is considered as the commonest studied variant in the human gene – which variant, which gene do you mean?

Introduction – is very confusing and redundant- need rewriting and focusing on the studied SNPs.

Line 177: normal menstrual cycles – why females with normal menses are excluded???

Line 183: EDTA blood – do you mean collecting blood on tubes with EDTA?

Line 184: In this study, the sample size calculation was followed by the previous study [56] in which we have included 105 cases and 105 controls. What do you mean? How is it followed by previous study? Do you mean adopted from previous study? – There is a sample size equation that should be calculated for each study, please calculate and provide references. Data used to calculate, should be from previous measurements to the polymorphism frequency rate or from a preliminary or pilot assessment.

Line 192: A couple of family histories and date of 192 birth were transformed into ages was recorded in both the cases and controls. The expertised women nurses were measured the BMI as well as blood was also drawn when visited to the outpatient clinic. – This is very confusing and need rewriting.

Line 208: Amplification of thrombophilic variants- do I really need to know the mins taken to complete the work and it equivalent hours!!!

Line 216: converted into 20ng/mL – are you sure it is / ml (1000μl) – This is very low!

Line 221: 72°C but varies – is it 72 or it varies??? – do you meant explaining the cycle extension and the final extension – please rewrite.

Line 222: 50μl reaction – do you mean / reaction.

Line 249: were found to be high in controls – which variable do you mean?

Line 245: Characteristics of the women population – what is the order of your group presentation – you mentioned the studied and then the control in the first line and then compared variables without clarifying which group vs the other and then in the comment it seems you meant control group what the first to mention – this is confusing – please specify and the control should be the first group to refer to.

Line 251: was also found to 251 be high in control women – please describe in terms of statistical significance.

Line 253: In this study, all FI cases were found to have 100% infertility and none of them were pregnant – are this is the inclusion criteria.

Line 348: To maximize power to detect the accuracy of heterozygous and variants present in the thrombophilic variants, we have opted the complete Saudi women towards this case-control 350 study and it can be considered as one of the strengths of this study – Could you please ckarify.

Line 387: Finally, if any of the control individ-387 ual develops homozygous variant in the Saudi population then it can be considered as disease causing variant in the future within their self-family as well as in the Saudi com-389 munity – based on what?

Line 399: One of the possible explanations for the contradictory results – where is contradictory results – the studies mentioned in Saudi Arabia and Arab countries are in agreement.

Discussion – a lot of unrelated information is included.

Line 435 – 442: The SNPs present in 3’UTR region on a gene- In our study, G20210A SNP was present in 3’UTR  – this is first time to discuss the nature of the studied SNPs – they should be detailed in the introduction whether they are in the coding sequence – intron or regulatory sequence and its impact if studied before – not only listed in the table.

Line 444 - exons present in SNPs – do you mean SNPs present in exons?

Line 445 - SNPs present in intron can attack RNA splicing – do you mean it can affect or span it?

Tables – where is footnotes - Table 7: GG=95  is those numbers represent the percentages?

Could you add astreik to significant results as in table 8.

Comments on the Quality of English Language

It needs extensive work to improve language.

Author Response

Reviewer comments 3

Comments and Suggestions for Authors

Abstract

- Line 29: among G1691A, G20210A and C677T SNPs among thrombophilia – please rephrase to remove one among.

Ans) The manuscript was edited with native experts.

- Line 38 to 43: I think it should be removed.

Ans) We have rectified the error.

- Line 56 to 59: not clear – please clarify the meaning.

Ans) We have updated in the revised manuscript.

- Line 59: One of the common causes of infertility in both the genders or couples could be dysfunction of the reproductive system [7] – repetitive.

Ans) We have deleted the repetition.

- Line 62: another formulation for the percentage.

Ans) We have updated in the revised manuscript.

- Please bundle all related information together and do not just repeat in different voice.

Ans) We have edited the manuscript and updated in the revised manuscript.

- Line 69 to 84: I did not get the relation of obesity to the studied polymorphism. Please be concise and to the point.

Ans) We have described the obesity in relation with female infertility in the Saudi women.

-Line 83 and 84: repetition.

Ans) We have revised the sentence in the revised manuscript.

- Line 116-124: What is the relation of study to obesity???

Ans) We have tried to explain a point in regards relation between female infertility and obesity. The prevalence of obesity in Saudi Arabia is growing up precisely in the women.

- Line 142: MTHFR is considered as the commonest studied variant in the human gene – which variant, which gene do you mean?

Ans) In this study, we have used C677T variant.

- Introduction – is very confusing and redundant- need rewriting and focusing on the studied SNPs.

Ans) Now, the manuscript was edited with the native experts and we have developed on the SNPs involved in this study.

- Line 177: normal menstrual cycles – why females with normal menses are excluded???

Ans)This is generally describing to include the control women and exclude the cases. Now, we have rectified the error.

- Line 183: EDTA blood – do you mean collecting blood on tubes with EDTA?

Ans) The peripheral blood was collected in an EDTA vacutainer to study the molecular analysis.

- Line 184: In this study, the sample size calculation was followed by the previous study [56] in which we have included 105 cases and 105 controls. What do you mean? How is it followed by previous study? Do you mean adopted from previous study? – There is a sample size equation that should be calculated for each study, please calculate and provide references. Data used to calculate, should be from previous measurements to the polymorphism frequency rate or from a preliminary or pilot assessment.

Ans) Our intension was to provide a reference after calculated and confirming a minimum of 98-105 samples in each group. However, in our study, we have included 100 cases and 100 controls. We have updated the sentence to avoid the confusion.

- Line 192: A couple of family histories and date of 192 birth were transformed into ages was recorded in both the cases and controls. The expertised women nurses were measured the BMI as well as blood was also drawn when visited to the outpatient clinic. – This is very confusing and need rewriting.

Ans) The manuscript was edited with the native experts and we have updated it.

- Line 208: Amplification of thrombophilic variants- do I really need to know the mins taken to complete the work and its equivalent hours!!!

Ans) We have tried to incorporate as one of the reviewers in my previous emails has recommended to include and I am kept on adding it all my publications.

-Line 216: converted into 20ng/mL – are you sure it is / ml (1000μl) – This is very low!

Ans) In general, we will be having around 100ng/mL concentration of the genomic DNA. We will be using 20 ng/mL to avoid the high DNA concentration. When we use 100ng/mL, the bands are very strong and very sharp and it will be difficult in analyzing the digested PCR products, when the band size is very low and based on this reason, we have optimized to 20ng/mL.

- Line 221: 72°C but varies – is it 72 or it varies??? – do you meant explaining the cycle extension and the final extension – please rewrite.

Ans) Both the extension and final extensions was carried at 72°C-45seconds and 72°C-5 minutes.

- Line 222: 50μl reaction – do you mean / reaction.

Ans) This is the PCR concentration carried out for 50µl

- Line 249: were found to be high in controls – which variable do you mean?

Ans) The mean of age, weight, height, BMI, LH levels and LH/FSH ratio was higher in controls when compared with the controls.

- Line 245: Characteristics of the women population – what is the order of your group presentation – you mentioned the studied and then the control in the first line and then compared variables without clarifying which group vs the other and then in the comment it seems you meant control group what the first to mention – this is confusing – please specify and the control should be the first group to refer to.

Ans) In this study, among the table-2, we have measured the demographic details between infertile (cases) women vs fertile (control) women. However, we have found age, weight, height, BMI, LH and LH/FSH ratio was high in fertile women and both the FSH and TSH levels were high in fertile women. Now, this paragraph was rewritten and edited the document for the better clarity.

- Line 251: was also found to 251 be high in control women – please describe in terms of statistical significance.

Ans) Now, we have added the p values

- Line 253: In this study, all FI cases were found to have 100% infertility and none of them were pregnant – are this is the inclusion criteria.

Ans) Yes, this is one of the inclusion criteria for female infertility cases.

- Line 348: To maximize power to detect the accuracy of heterozygous and variants present in the thrombophilic variants, we have opted the complete Saudi women towards this case-control 350 study and it can be considered as one of the strengths of this study – Could you please clarify.

Ans) Our intension was to convey as to detect the accurate genotypes (heterozygous and homozygous variants) among the female infertile women, we have selected only Saudi women rather mixed women population in Saudi Arabia which means, we haven’t involved non-Saudi infertile women.

- Line 387: Finally, if any of the control individ-387 ual develops homozygous variant in the Saudi population then it can be considered as disease causing variant in the future within their self-family as well as in the Saudi com-389 munity – based on what?

Ans) In this study, none of the control women developed homozygous variant in any of the 3 SNPs of the thrombophilic genes and if any of the control women develops the homozygous variant, then that women can be considered disease causing variant in the future. This sentence was recommended based on the obtained genotype data from our study.

- Line 399: One of the possible explanations for the contradictory results – where is contradictory results – the studies mentioned in Saudi Arabia and Arab countries are in agreement.

Ans) The contradictory results mean inconsistent/non-significant (negative) results.

- Discussion – a lot of unrelated information is included.

Ans) We have worked on the following comments in the revised manuscript.

- Line 435 – 442: The SNPs present in 3’UTR region on a gene- In our study, G20210A SNP was present in 3’UTR – this is first time to discuss the nature of the studied SNPs – they should be detailed in the introduction whether they are in the coding sequence – intron or regulatory sequence and its impact if studied before – not only listed in the table.

Ans) In the introduction, we have added a sentence regarding the presence of SNPs in a specific region. We want to continue the above sentences in the discussion as we’re maintain a flow with the SNPs of thrombophilic genes. I hope you will allow us to do it.

- Line 444 - exons present in SNPs – do you mean SNPs present in exons?

Ans) Both G1691A and C677T SNPs were present in the exonic region.

- Line 445 - SNPs present in intron can attack RNA splicing – do you mean it can affect or span it?

Ans) The SNPs present in the introns can have RNA splicing which means it is a process of conversion of DNA into RNA and the genetic information can be modified. There is a probability of effecting.

Tables – where is footnotes - Table 7: GG=95 is those numbers represent the percentages?

Ans) Now, we have added the footnotes and in Table 7, GG=95 is both the numbers as well as percentages. However, we have indicated as the total number and it can also represent the percentage.

Could you add astreik to significant results as in table 8.

Ans) Now, we have added the asterisk to the p value with the presence of the significant results

Reviewer 4 Report

Comments and Suggestions for Authors

The research topic is well chosen; however, the research results are insufficient to write an article. I agree that there can be studies with negative results, but still, something more needs to be done. You have done nothing but demonstrate that the hypothesis of the study does not hold up.

The manuscript does not seem very well prepared early on as you have not even removed the instructions from the journal template. The article has serious editing/expression problems in English. Some of these lead to the complete loss of the statement's meaning.  You are using the phrase "In this study" for almost every sub-section of the article, please revise this.

In the Abstract you state that in this study you would like to investigate (L 20). Haven't you already?

The method description is faulty. For example, describe how much time you spent in the lab isolating DNA. This is seriously wrong; it cannot be presented in a scientific article. What must be added here are the bibliographic sources (for the chosen method, primers, etc.).

Unfortunately, the Tables where you claim to present the research results were not made available to me. For a precise review, I also need this data.

Comments on the Quality of English Language

The quality of the English language in this manuscript is very seriously affected. I didn't spot a lot of misspellings, but there were a lot of statements that didn't make sense. Also, some words are wrong (i.e. liver instead of alive). From this point of view, the manuscript requires a serious re-editing.

Author Response

Reviewer comments 4

Comments and Suggestions for Authors

The research topic is well chosen; however, the research results are insufficient to write an article. I agree that there can be studies with negative results, but still, something more needs to be done. You have done nothing but demonstrate that the hypothesis of the study does not hold up.

Ans) Dear Reviewer, thank you for your valuable comment. The concept of this study was to screen the minimum of 3 thrombophilic variants in female infertility patients. Unfortunately, we have confirmed the negative association. However, the cons of this study were discussed in the revised manuscript and edited with the native experts.

The manuscript does not seem very well prepared early on as you have not even removed the instructions from the journal template. The article has serious editing/expression problems in English. Some of these lead to the complete loss of the statement's meaning.  You are using the phrase "In this study" for almost every sub-section of the article, please revise this.

Ans)The manuscript was edited with the native experts. We apologize for it.

In the Abstract you state that in this study you would like to investigate (L 20). Haven't you already?

Ans) The manuscript was prepared based on the issue of plagiarism and now, we have edited the manuscript with the native experts.

The method description is faulty. For example, describe how much time you spent in the lab isolating DNA. This is seriously wrong; it cannot be presented in a scientific article. What must be added here are the bibliographic sources (for the chosen method, primers, etc.).

Ans) Now, we have updated in the revised manuscript. We have tried differently to present it but now, we have rectified it. Thank you for your comment. We really appreciate it.

Unfortunately, the Tables where you claim to present the research results were not made available to me. For a precise review, I also need this data.

Ans) All the data was present in the tables. If you need raw data, kindly let me know. We can provide it.

Comments on the Quality of English Language

The quality of the English language in this manuscript is very seriously affected. I didn't spot a lot of misspellings, but there were a lot of statements that didn't make sense. Also, some words are wrong (i.e., liver instead of alive). From this point of view, the manuscript requires a serious re-editing.

Ans) Now, the manuscript was edited with the native experts and updated in the revised manuscript.

Round 2

Reviewer 1 Report

Comments and Suggestions for Authors

The manuscript has been revised accordingly and most of the issues were addressed.

Author Response

Reviewer comments 1

The manuscript required major correction

  1. There are a lot of errors in the manuscript that need to be corrected. The grammar needs to be improved.
  2. A) The manuscript was edited the native experts and rectified the grammatical errors.
  3. Abstract: The abstract should be clear and concise, stating the method used, findings, and conclusion.
  4. A) We have edited the manuscript and updated the abstract as per your suggestion.
  5. Line 38 - 43: Are they relevant? Pl, remove.
  6. A) We have removed it.
  7. There are a lot of inconsistencies in the manuscript for example in the abstract the population size mentioned was 200. However, in the methodology section 210 and 220 population sizes were mentioned.
  8. A) We have rectified the error. In this study, we have used 100 cases and 100 controls.
  9. The rs numbers mentioned should be uniform throughout the manuscript. The rs numbers mentioned in the discussion section were different from those reported in the tables.
  10. A) Now, we have updated in the revised manuscript.
  11. No consistency in the unit format: The Format of writing units should be uniform all through
  12. A) The manuscript was edited with the native experts and revised and updated in the revised manuscript.
  13. The discussion needs to be improved
  14. A) We have done our best in developing the discussion.

Reviewer 2 Report

Comments and Suggestions for Authors

Although the authors tried rectifying the errors, in my opinion the study still doesn't have substantial merit to be published in its present form.

Comments on the Quality of English Language

The English language has been corrected

Author Response

Reviewer comments 2

Comments and Suggestions for Authors

In the article entitled "Molecular screening of thrombophilic variants among Saudi infertile women", Alageel, A.A. et al. aims to explore the association between G1691A, G20210A, and C677T variants of thrombophilia in FVL, FII and MTHFR genes among Saudi infertile women. The reviewer has the following comments:

  1. Although the study holds some merit, in my opinion it is short of being a significant contribution to the field. The rationale behind some of the analyses are not clearly explained. The narrative needs to be significantly improved.
  2. A) Dear Reviewer, thank you for your valuable comments. The manuscript was edited with the native experts and updated in the revised manuscript.

  3. Based on what did the authors choose the three genes in the study among a total of 13 thrombophilic genes?  Also, did the choice of the genes influence the final conclusion that thrombophilic SNPs have no role and involvement in infertility?
  4. A) Among 13 thrombophilic genes, we have opted only 3 genes. The selection was based on its role in the Saudi population and we have confirmed selection of single SNP from 1 gene and selection of only 3 genes was one of the limitations of our study.
  5. For some of the result sections, no clear rationale or conclusions is provided summarizing the findings. A summary statement would greatly enhance readability and clarity.
  6. A) Now, the manuscript was edited with the native experts and updated in the revised manuscript. Now, we assume that we have justified the raised query.

Comments on the Quality of English Language

The quality of English language must be significantly improved to enhance the clarity of the article. Sometimes it is difficult to comprehend the meaning of the sentences. The article contains numerous typos and grammatical errors which must be corrected. 

A) The manuscript was edited with the native experts and both typos and grammatical errors were rectified.

Reviewer 3 Report

Comments and Suggestions for Authors

Dear authors,

The research has a valid question, but it still needs language improvement. I have pointed out to some scientific comments that are not addressed or the reply did not provide a valid rationale as - methods: amplification of variants: you mentioned ‘DNA samples were converted into 20 ng/mL’, you replied you tried 100 and the band was dense. But honestly even 100/ml is less for the reaction. We are using X/ul to measure DNA concentration having it to mls and it is two digit number will probably not yield enough amplicon.

Again in the results under Characteristics of the women : I mentioned before that the order you mentioned the cases and controls did not match the following results and it is not corrected!!!

Discussion: I do not want to go through all the language edits again but there are some words that not only confusing or misleading but further it is not scientifically sound like ‘the exons present in SNPs’ (I mentioned it before – now it is line 430) How and an exon is present in the SINGLE nucleotide polymorphism.

I would advise that before publication the work needs to be revised word by word by two experts in the field and then checked by a native speaker. Wishing you good luck!

Comments on the Quality of English Language

More improvement please.

Author Response

Reviewer comments

The research has a valid question, but it still needs language improvement. I have pointed out to some scientific comments that are not addressed or the reply did not provide a valid rationale as - methods: amplification of variants: you mentioned ‘DNA samples were converted into 20 ng/mL’, you replied you tried 100 and the band was dense. But honestly even 100/ml is less for the reaction. We are using X/ul to measure DNA concentration having it to mls and it is two-digit number will probably not yield enough amplicon.

A) Dear Reviewer, Thank you for your question. First of all, we have re-edited the manuscript. Next, we convert our DNA into 20ng/mL (A260/A280 ratio was between 1.8-2.0) and it has worked out for all our publications. You can refer my previous publications and we have mentioned the same amount of concentration.

Again, in the results under Characteristics of the women: I mentioned before that the order you mentioned the cases and controls did not match the following results and it is not corrected!!!

A) We apologize for this error. Now, we have updated in the revised manuscript.

Discussion: I do not want to go through all the language edits again but there are some words that not only confusing or misleading but further it is not scientifically sound like ‘the exons present in SNPs’ (I mentioned it before – now it is line 430) How and an exon is present in the SINGLE nucleotide polymorphism.

A) We have again edited the manuscript. Our intension was to convey as SNPs present in the exon.

I would advise that before publication the work needs to be revised word by word by two experts in the field and then checked by a native speaker. Wishing you good luck!

A) Arwa and Dr. Maysoon has rechecked the complete manuscript, edited and rectified the errors.
